# Fast Unbalanced Optimal Transport on a Tree

**Ryoma Sato**[1,2]    **Makoto Yamada**[1,2]    **Hisashi Kashima**[1,2]
[1]Kyoto University    [2]RIKEN AIP
{r.sato@ml.ist.i, myamada@i, kashima@i}.kyoto-u.ac.jp

## Abstract

This study examines the time complexities of the unbalanced optimal transport problems from an algorithmic perspective for the first time. We reveal which problems in unbalanced optimal transport can/cannot be solved efficiently. Specifically, we prove that the Kantorovich Rubinstein distance and optimal partial transport in the Euclidean metric cannot be computed in strongly subquadratic time under the strong exponential time hypothesis. Then, we propose an algorithm that solves a more general unbalanced optimal transport problem exactly in quasi-linear time on a tree metric. The proposed algorithm processes a tree with one million nodes in less than one second. Our analysis forms a foundation for the theoretical study of unbalanced optimal transport algorithms and opens the door to the applications of unbalanced optimal transport to million-scale datasets.

## 1 Introduction

The optimal transport (OT) distance is an effective tool to compare measures and is used in a variety of fields. The applications of OT include image processing [45, 50, 23], natural language processing [38, 52], biology [57, 43, 26], and generative models [4, 54]. However, one of the major limitations of OT is that it cannot handle measures with different total mass.

We illustrate two specific issues here. First, the amount of mass is an important signal, e.g., in shape analysis [58, 17] and comparing persistence diagrams [39, 61]. Secondly, the OT distance is susceptible to noise because we must transport noise mass with high costs if noise occurs far away from other mass. In many cases, the measures are normalized so that they become probabilistic measures for applying the OT distance to them. However, normalization loses the information of the amount of mass and does not solve the noise problem, as illustrated in Figure 1, where blue and red bars represent two input measures, and the gray bar represents destructed mass.

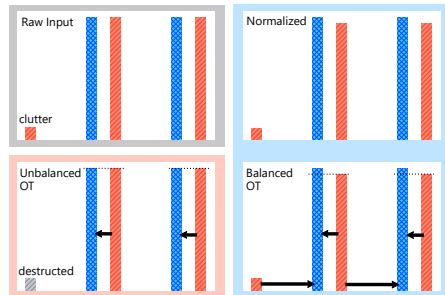

Figure 1: Illustrative example.

To overcome these issues, many variants of the OT distance in the unbalanced setting have been proposed, such as the optimal partial transport [12, 27] and Kantorovich Rubinstein distance [41, 31]. We first propose a unified framework to discuss these problems as the generalized Kantorovich Rubinstein (GKR) distance, which encompasses many existing unbalanced OT distances.

$$\min_{\pi \in \mathcal{U}^-(\mu,\nu)} \int_{\mathcal{X} \times \mathcal{X}} c(x,y) d\pi(x,y) + \int_{\mathcal{X}} \lambda_d(x) d(\mu - \mathrm{proj}_1 \pi)(x) + \int_{\mathcal{X}} \lambda_c(y) d(\nu - \mathrm{proj}_2 \pi)(y).$$

The formal notations will be provided later. Intuitively, the GKR distance can destruct a unit mass at cost $\lambda_d(x)$ and create a unit mass at cost $\lambda_c(x)$ at $x \in \mathcal{X}$. Therefore, $\lambda_c = \lambda_d = \infty$ recovers

the standard OT distance. We investigate the GKR distance in the light of fine-grained complexity [62, 63]. Fine-grained complexity shows that some problem is not solvable within $O(n^{c-\varepsilon})$ time under some hypothesis, just like NP-hard problems are shown to be not solvable efficiently under the P $\neq$ NP hypothesis. In this paper, we prove that important special cases of the GKR distance, namely the optimal partial transport and Kantorovich Rubinstein distance, are not solvable in strongly subquadratic time under the strong exponential time hypothesis (SETH) [32, 63].

Thanks to this theorem, we can avoid fruitless efforts pursuing efficient algorithms for unbalanced OT in the Euclidean space, and this theorem motivates us to consider other spaces. It is known that the OT distance in the Euclidean space can be efficiently approximated by the OT distance in the 1-dimensional Euclidean space [50, 36, 13] or tree metrics [33, 14, 40]. We consider the GKR distance on tree metrics in this paper. It is noteworthy that tree metrics include 1-dimensional space because 1-dimensional space can be seen as a path graph, which is a special case of a tree. We prove that the GKR distance on a quadtree metric can approximate the GKR distance on the Euclidean metric theoretically and empirically. Although it is easy to compute the OT distance in tree metrics in linear time, it is not trivial how to compute the GKR distance in tree metrics efficiently. In this paper, we propose an efficient algorithm to compute the GKR distance in tree metrics in $O(n \log^2 n)$ time, where $n$ is the number of nodes in a tree. Therefore, our algorithm solves many existing unbalanced OT problems on tree metrics efficiently. In practice, our algorithm processes a tree with more than one million nodes in less than one second on a laptop computer. Our code including a C++ implementation and Python wrapper for our algorithm is available at `https://github.com/joisino/treegkr`.

The contributions of this paper are as follows. **Framework:** We propose the GKR distance, a general framework for unbalanced OT problems. **Hardness result:** We prove that existing unbalanced OT problems cannot be solved in strongly subquadratic time under SETH. **Efficient algorithm:** We propose a quasi-linear time algorithm for the GKR distance in tree metrics.

## 2 Related Work

One of the major constraints of the OT distance is that it requires the two input measures to have the same total mass. However, handling measures with different mass is crucial in imaging [41, 18, 17], biology [57], keypoint matching [55], text matching [59], generative models [64], and transport equations [48, 49]. Kantorovich [35, 34] already extended the OT distance to the unbalanced setting by introducing the waste function in 1957. Benamou [10, 8] generalized the OT distance to the unbalanced setting via the dynamic formulation of the OT distance [9]. Following this work, many generalizations of the OT distance to the unbalanced setting have been proposed, such as the optimal partial transport [12, 27], Wasserstein-Fisher-Rao [42, 37, 17], and Kantorovich-Rubinstein distance [41, 31]. Chizat et al. provided a unified view to the static and dynamic formulations of unbalanced OT [19] and proposed a generalized Sinkhorn algorithm to compute unbalanced OT efficiently [18].

Sliced partial optimal transport [11] is a linear time algorithm for a special case of the optimal partial transport in the 1-dimensional Euclidean space. Their problem is a very special case of the generalized Kantorovich Rubinstein distance because the GKR distance with $\lambda_c = 0, \lambda_d = \infty$ recovers their case. Moreover, we propose an algorithm for tree metrics, which can handle 1-dimensional space (i.e., a path graph) as a special case.

Lellmann et al. [41] utilized the Kantorovich Rubinstein distance, where the cost of destruction and creation is uniform (i.e., $\lambda_c = \lambda_d = \lambda$ (const.)), for denoising and cartoon-texture decomposition. Uniform destruction costs (i.e., adding dustbins) are used in keypoint matching [24, 55] and text matching [59] as well to absorb unmatched points. Caffarelli et al. [12] and Figalli [27] proposed optimal partial transport to handle unbalanced measures. This metric transports $\kappa \leq \min(\|\mu\|_1, \|\nu\|_1)$ mass instead of all mass. As Chizat et al. [17] pointed out, this is equivalent to the Kantorovich Rubinstein distance. This indicates that the earth mover's distance (EMD) [53] and Pele's EMD [47] are also special cases of GKR (with additional trivial terms or normalization).

Kantorovich considered a variant of the OT distance by allowing mass creation and destruction at the boundary of the domain [35]. Figalli et al. [28] considered a similar distance and an application to gradient flows with Dirichlet boundary conditions. Their distance is a special case of the GKR distance because GKR with $\lambda_c(x) = \lambda_d(x) = d(x, \partial\mathcal{X})$ recovers their distance, where $d(x, \partial\mathcal{X})$ is the distance from $x$ to the boundary. Lacombe et al. [39] used the unbalanced OT distance for comparing persistent diagrams. They considered the diagonal of persistent diagrams as a sink. Their

distance is a special case of the GKR distance because GKR with $\lambda_c(x) = \lambda_d(x) = d(x, \Delta)$ recovers their problem, where $d(x, \Delta)$ is the distance from $x$ to the diagonal.

A network flow-based method [31, 46] can compute the GKR distance, but a major limitation of the flow-based method is its scalability. Namely, the flow-based method runs in $O(n^2 \log n)$ time even on tree metrics. In this paper, we propose an efficient algorithm to compute the GKR distance in tree metrics exactly that works in $O(n \log^2 n)$ time in the worst case. Our method can process measures with more than one million elements within one second.

Some existing works proposed fast computation of *standard* OT [30, 3]. For example, Genevay et al. [30] reported that their algorithm processed measures with 20000 points in less than two minutes with Tesla K80 cards. Our algorithm computes *unbalanced* OT distances between two measures with *one million* points in less than *one second* with *one CPU core*, which is much faster than existing algorithms. Besides, it should be noted that although special cases of standard OT, such as OT on trees and Euclidean spaces, can be computed efficiently [3, 40], and unbalanced OT can be converted to standard OT [31], the converted OT problems are not necessarily in Euclidean (resp. tree) spaces even if the original unbalanced OT problems are in Euclidean (resp. tree) spaces. Thus, such methods are not necessarily applicable to unbalanced OT problems directly.

**Limitation of our framework:** The GKR distance penalizes mass creation and destruction linearly. Thus, the GKR distance does not contain the Wasserstein-Fisher-Rao distance [42, 37, 17], which penalizes mass creation and destruction by KL divergence. Extending our results to the Wasserstein-Fisher-Rao distance is an important open problem.

## 3 Background

**Notations.** $\mathcal{M}(\mathcal{X})$ denotes the set of measures on measurable space $\mathcal{X}$. When the measurable space $\mathcal{X} = \{x_1, \ldots, x_n\}$ is finite, a measure $\mu = \sum_i a_i \delta_{x_i}$ can be represented as a histogram $\boldsymbol{a} = [a_1, \ldots, a_n]^\top \in \mathbb{R}^n_{\geq 0}$, where $\delta_x$ is the Dirac mass at $x$. We use a measure and a histogram interchangeably in that case. $\text{proj}_1$ and $\text{proj}_2 \colon \mathcal{M}(\mathcal{X} \times \mathcal{X}) \to \mathcal{M}(\mathcal{X})$ are projections to the first and second coordinate, respectively. Specifically, for a measure $\mu \in \mathcal{M}(\mathcal{X} \times \mathcal{X})$ and measurable sets $A, B \subset \mathcal{X}$, $\text{proj}_1 \mu(A) = \mu(A \times \mathcal{X})$ and $\text{proj}_2 \mu(B) = \mu(\mathcal{X} \times B)$. $\mathcal{U}(\mu, \nu) = \{\pi \in \mathcal{M}(\mathcal{X} \times \mathcal{X}) \mid \text{proj}_1 \pi = \mu, \text{proj}_2 \pi = \nu\}$ denotes the set of coupling measures of $\mu$ and $\nu \in \mathcal{M}(\mathcal{X})$. $\mathcal{U}^-(\mu, \nu) = \{\pi \in \mathcal{M}(\mathcal{X} \times \mathcal{X}) \mid \text{proj}_1 \pi \leq \mu, \text{proj}_2 \pi \leq \nu\}$ denotes the set of sub-coupling measures of $\mu$ and $\nu \in \mathcal{M}(\mathcal{X})$. A tree is a connected acyclic graph. In this paper, we consider weighted undirected graphs. Thus, a tree is represented as a tuple $\mathcal{T} = (\mathcal{X}, E, w)$, where $\mathcal{X}$ is a set of nodes, $E \subset \mathcal{X} \times \mathcal{X}$ is a set of edges, and $w \colon E \to \mathbb{R}_{\geq 0}$ is a weight function. Because we consider undirected trees, if $(x, y) \in E$, $(y, x) \in E$ and $w(x, y) = w(y, x)$ hold. $d_\mathcal{T} \colon \mathcal{X} \times \mathcal{X} \to \mathbb{R}_{\geq 0}$ denotes the geodesic distance between two nodes on tree $\mathcal{T}$. Specifically, for node $u, v \in \mathcal{X}$, $d_\mathcal{T}(u, v)$ is the sum of the edge weights in the unique path between $u$ and $v$. For $f, g \colon \mathbb{R} \to \mathbb{R}$, $f = \omega(g)$ denotes $f(x)/g(x) \to \infty$ as $x \to \infty$.

**Optimal Transport.** Given two measures $\mu$ and $\nu \in \mathcal{M}(\mathcal{X})$ with the same total mass (i.e., $\|\mu\|_1 = \|\nu\|_1$) and a cost function $c \colon \mathcal{X} \times \mathcal{X} \to \mathbb{R}_{\geq 0}$, the OT problem is defined as $\text{OT}(\mu, \nu) = \min_{\pi \in \mathcal{U}(\mu, \nu)} \int_{\mathcal{X} \times \mathcal{X}} c(x, y) d\pi(x, y)$. In particular, when the cost function is the power $d^p$ of a distance $d$, $\text{OT}^{1/p}$ is referred to as the Wasserstein distance. The Wasserstein distance has many applications in machine learning, including document classification [38], comparing label distributions [29], and generative models [4, 54]. The OT problem can be solved using a minimum cost flow algorithm [46] exactly or using the Sinkhorn algorithm [22] with entropic regularization. One limitation of the OT problem is that it cannot handle measures with different total mass because the set $\mathcal{U}(\mu, \nu)$ of coupling measures is empty in that case. Many extensions of the OT problem have been proposed to deal with "unbalanced" measures, as we reviewed in Section 2. In the following sections, we analyze the time complexities of unbalanced OT problems.

## 4 Generalized Kantorovich Rubinstein Distance

In this section, we propose the generalized Kantorovich Rubinstein (GKR) distance, a generalized problem of unbalanced OT. The GKR distance is defined as follows:

$$\text{GKR}(\mu, \nu) = \min_{\pi \in \mathcal{U}^-(\mu,\nu)} \int_{\mathcal{X} \times \mathcal{X}} c d\pi + \int_{\mathcal{X}} \lambda_d d(\mu - \text{proj}_1 \pi) + \int_{\mathcal{X}} \lambda_c d(\nu - \text{proj}_2 \pi),$$

where $\lambda_d$ and $\lambda_c \colon \mathcal{X} \to \mathbb{R}_{\geq 0}$ are destruction and creation cost functions, respectively. Intuitively, the GKR distance does not necessarily transport all the mass but pays penalties for mass creation and destruction. It should be noted that the GKR distances are so general that the metric axioms do not always hold. The triangle inequality holds if (1) $c$ is a metric and (2) $\lambda_d(x) \leq c(x,y) + \lambda_d(y)$ and $\lambda_c(y) \leq \lambda_c(x) + c(x,y)$ hold for any $x, y \in \mathcal{X}$. We assume condition (2) in the following. Intuitively, this condition says that it is always more beneficial to destruct a mass at $x$ than to transport the mass somewhere and destruct it there. When $c$ is symmetric and $\lambda_d = \lambda_c$, the GKR is also symmetric by its definition. Note that we do assume only condition (2) in the following, thus our propose method can be used for not only special cases of GKR where the metric axioms hold but also non-metric GKR distances. Importantly, the GKR distance includes many popular variants of the OT distance as special cases, as we discussed in Section 2. namely, the GKR distance encompasses the Kantorovich Rubinstein distance [41, 31], optimal partial transport problem [12, 27], and Figalli's formulation [28, 35] as special cases.

Measures considered in the machine learning field are often discrete and endowed with the Euclidean metric [38, 21]. For the time being, we consider that the space is a finite subset of the $d$-dimensional space (i.e., $\mathcal{X} \subset \mathbb{R}^d$ and $|\mathcal{X}| = n$), and the cost function is the power of the $L_p$ metric (i.e., $c(x,y) = \|x - y\|_p^p$), and we show that important special cases of the GKR distance cannot be computed efficiently under the following hypothesis.

**Hypothesis 1** (strong exponential time hypothesis (SETH) [32, 63]). For any $\delta < 1$, there exists $k \in \mathbb{Z}^+$ such that the $k$-SAT problem with $n$ variables cannot be solved in $O(2^{\delta n})$ time even by a randomized algorithm.

This hypothesis has been used to prove that some problems, such as graph diameter [51, 15] and edit distance [6], are not solvable efficiently and has been supported by, e.g., [1, 16]. We show that unbalanced optimal transport problems cannot be solved efficiently under this hypothesis.

**Theorem 2.** *If SETH is true, for any $p \geq 1$ and $\varepsilon > 0$, neither the Kantorovich Rubinstein distance nor optimal partial transport problem, where $\mathcal{X} \subset \mathbb{R}^d$, $c(x,y) = \|x - y\|_p^p$, and $d = \omega(\log n)$, can be solved in $O(n^{2-\varepsilon})$ time.*

This theorem is proved by reduction from the bichromatic Hamming close pair problem [2]. All proofs are provided in the appendices. As far as we know, fine grained complexity has not yet been explored in the machine learning literature, except empirical risk minimization [7]. Our result demonstrates that the concept of fine grained complexity is a useful tool to derive hardness results in machine learning. From this theorem, it seems impossible to apply unbalanced OT to million-scale datasets in the Euclidean space. This fact motivates us to consider easier metrics. Previous works proved that OT in the Euclidean space can be effectively approximated by OT on a tree metric via theoretical arguments [14, 33] and empirical studies [33, 40]. In the quadtree approximation, the ground cost between two nodes is defined as the distance between the centers of regions. It should be noted that the quadtree and its theory are not limited to two dimensions, though too high dimensions are prohibited due to its scalability. We extend the results of the quadtree OT approximation by Indyk and Thaper [33] to the tree GKR approximation.

**Theorem 3.** *Let $GKR_{euc}$ be the GKR distance with Euclidean cost $c_{euc}(x,y) = \|x-y\|_2$. Let $GKR_{tree}$ be the GKR distance with quadtree cost $c_{tree}(x,y) = d_{\mathcal{T}}(x,y)$, where $\mathcal{T}$ is a quadtree. There exists a constant $C$ such that for any measures $\mu$ and $\nu$, $GKR_{euc}(\mu, \nu) \leq C \cdot GKR_{tree}(\mu, \nu)$ holds. If we randomly translate measures when we construct a quadtree, there exists a constant $D$ such that $\mathbb{E}_{\mathcal{T}}[GKR_{tree}(\mu, \nu)] \leq D \cdot GKR_{euc}(\mu, \nu) \log \Delta$, where $\Delta$ is the spread (i.e., the ratio of the farthest distance to the closest distance), and the expectation is taken by random translation of the measures.*

Note that $C$ and $D$ are dependent on $d$ but independent of $n$. This theorem indicates that the GKR distance on the Euclidean metric can be approximated by the GKR distance on a quadtree metric. Besides, tree OT includes the sliced OT [50, 36, 13], which was shown to be an effective approach to scale up the standard OT. In addition to that, OT on a tree metric is interesting in its own right. For example, UniFrac [43] uses OT on a phylogenetic tree to compare microbial communities. In the following, we consider GKR on a tree metric. Formally, the problem of the GKR distance on a tree metric can be formalized as follows.

**Problem 4** (GKR distance on tree metrics). **Input:** A tree $\mathcal{T} = (\mathcal{X}, E, w)$ with $n = |\mathcal{X}|$ nodes, mass destruction and creation functions $\lambda_d, \lambda_c \colon \mathcal{X} \to \mathbb{R}_{\geq 0}$, and two measures $\boldsymbol{a}, \boldsymbol{b} \in \mathbb{R}_{\geq 0}^{\mathcal{X}}$ on tree $\mathcal{T}$. **Output:** The GKR distance $\mathrm{GKR}(\boldsymbol{a}, \boldsymbol{b})$ with cost $c(x, y) = d_{\mathcal{T}}(x, y)$.

Note that the GKR distance on a finite space can be formulated as follows.

$$\mathrm{GKR}(\boldsymbol{a}, \boldsymbol{b}) = \min_{\pi \in \mathcal{U}^-(\boldsymbol{a}, \boldsymbol{b})} \sum_{x, y \in \mathcal{X}} c(x, y) \pi_{x,y} + \sum_{x \in \mathcal{X}} \lambda_d(x)(\boldsymbol{a} - \mathrm{proj}_1 \pi)_x + \sum_{y \in \mathcal{X}} \lambda_c(y)(\boldsymbol{b} - \mathrm{proj}_2 \pi)_y,$$

where $(\mathrm{proj}_1 \pi)_i = \sum_j \pi_{ij}$ and $(\mathrm{proj}_2 \pi)_j = \sum_i \pi_{ij}$ are projections and $\mathcal{U}^-(\boldsymbol{a}, \boldsymbol{b}) = \{\pi \in \mathbb{R}_{\geq 0}^{\mathcal{X} \times \mathcal{X}} \mid (\mathrm{proj}_1 \pi)_x \leq \boldsymbol{a}_x, (\mathrm{proj}_2 \pi)_y \leq \boldsymbol{b}_y\}$ is the set of sub-couplings. This problem can be solved by a minimum cost flow algorithm, as Guittet [31] pointed out for the Kantorovich Rubinstein distance. Specifically, the space is extended with a virtual point, and each point is connected to this point with costs of mass destruction and creation. Because there are $O(n)$ edges, this problem can be solved in $O(n^2 \log n)$ time by Orlin's algorithm [46]. The Sinkhorn algorithm [22] is an alternative approach to solve this problem in $O(n^2)$ time by introducing entropic regularization. However, these algorithms are too slow for large datasets. In the following section, we propose a more efficient algorithm for this problem.

# 5 Fast Computation of GKR on Tree Metrics

In this section, we propose an efficient algorithm for the GKR distance on tree metrics based on dynamic programming and speed up the computation using fast convex min-sum convolution, efficient data structures, and weighted-union heuristics[1]. For the algorithm description, we arbitrarily choose a root node $r \in \mathcal{X}$. Further, without loss of generality, we assume that the input is a binary tree, only leaf nodes have mass (i.e., $\boldsymbol{a}_x = \boldsymbol{b}_x = 0$ for all internal node $x$), and we do not create or destruct mass in internal nodes to simplify the discussion (see Appendix D).

**Summary.** The dynamic programming computes the GKR distance from leaf to root recursively. Figure 5 shows examples. Note that when we compute the transport in a parent node, the optimal assignments in the children subtrees are already computed recursively. When two children are merged, the dynamic programming determines the optimal transition (i.e., the optimal amount of mass that are transported to the left and right children). The naive implementation searches the transition of minimum cost from the all possibilities (i.e., the min operator in Eq. (3)), which leads to a slow computation. The proposed fast convolution algorithm speeds up this merge operation.

**Notations for algorithm description.** For a rooted tree $\mathcal{T} = (\mathcal{X}, E, w)$ and $v \in \mathcal{X}$, let $\mathcal{T}(v) = (\mathcal{X}_v, E_v, w)$ be the subtree of node $v$. Let $p(v) \in \mathcal{X}$ be the parent node of $v \in \mathcal{X}$. For a non-root node $v \in \mathcal{X}$, let $\mathcal{T}^+(v) = (\mathcal{X}_v \cup \{p(v)\}, E_v \cup \{(v, p(v)), (p(v), v)\}, w)$ be the extended subtree of node $v$. For a subtree $\mathcal{T}' = (\mathcal{X}', E', w)$ and measure $\boldsymbol{a} \in \mathbb{R}_{\geq 0}^{\mathcal{X}}$ on tree $\mathcal{T}$, let $\boldsymbol{a}|_{\mathcal{T}'} \in \mathbb{R}_{\geq 0}^{\mathcal{X}'}$ be the restriction of $\boldsymbol{a}$ to subtree $\mathcal{T}'$ (i.e., $\boldsymbol{a}|_{\mathcal{T}', v} = \boldsymbol{a}_v$ for $v \in \mathcal{X}'$). For $x \in \mathbb{R}$, let $[x]_+ = \max(0, x)$. For real-valued functions $f$ and $g \colon \mathbb{R} \to \mathbb{R}$, let $f * g \colon \mathbb{R} \to \mathbb{R}$ be the min-sum convolution of $f$ and $g$, i.e., $(f * g)(x) = \min_z f(x - z) + g(z)$.

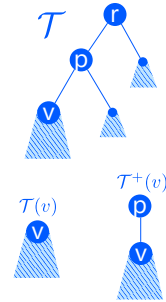

**Naive dynamic programming.** For each $v \in \mathcal{X}, x \in \mathbb{R}$, we consider the following two states:

$$t_{v,x} = \mathrm{GKR}(\boldsymbol{a}|_{\mathcal{T}(v)} + [x]_+ \cdot \delta_v, \boldsymbol{b}|_{\mathcal{T}(v)} + [-x]_+ \cdot \delta_v),$$

$$e_{v,x} = \mathrm{GKR}(\boldsymbol{a}|_{\mathcal{T}^+(v)} + [x]_+ \cdot \delta_{p(v)}, \boldsymbol{b}|_{\mathcal{T}^+(v)} + [-x]_+ \cdot \delta_{p(v)}).$$

Intuitively, $t_{v,x}$ and $e_{v,x}$ are restrictions of the GKR distance to subtrees $\mathcal{T}(v)$ and $\mathcal{T}^+(v)$, respectively, with additional $x$ mass on $v$ and $p(v)$, respectively. Therefore, the answer we want is $\mathrm{GKR}(\boldsymbol{a}, \boldsymbol{b}) = t_{r,0}$. We explain how to compute these values recursively. **Initial value of $t_v$:** In a leaf node $v$, the only thing we can do is create and destroy mass to balance the source and target mass at

[1] A similar technique is known in the competitive programming community `https://icpc.kattis.com/problems/conquertheworld`

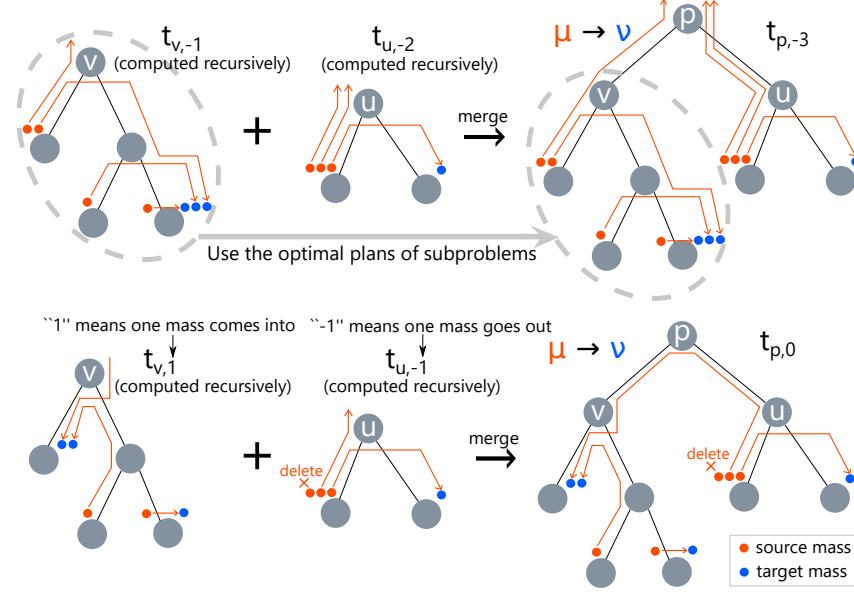

Figure 2: **Illustration of the DP computation.** The dynamic programming computes assignments recursively from leaf to root.

$v$. Therefore, the initial states in a leaf node $v$ are computed as follows:

$$t_{v,x} = \begin{cases} (\boldsymbol{b}_v - \boldsymbol{a}_v - x)\lambda_c(v) & (\boldsymbol{b}_v - \boldsymbol{a}_v - x \geq 0) \\ (\boldsymbol{a}_v + x - \boldsymbol{b}_v)\lambda_d(v) & (\text{otherwise}) \end{cases} \tag{1}$$

**Recursive equation of $e_v$:** For each non-root node $v$, the only thing we can do when we extend the subtree is transporting all mass on $p(v)$ to $v$ or $v$ to $p(v)$. Transporting $x$ mass costs $|x| \cdot w(v, p(v))$.
**Recursive equation of $t_v$:** For each internal node $p$ with children $v$ and $u$, when we merge two extended subtrees $\mathcal{T}'(v)$ and $\mathcal{T}'(u)$, the mass on $p$ are distributed to two extended subtrees so that the total cost is minimized. We search the distributed mass $y$ to child $u$ naively. Therefore, the following recursive equations hold:

$$e_{v,x} = |x| \cdot w(v, p(v)) + t_{v,x}, \tag{2}$$

$$t_{p,x} = \min_y e_{v,x-y} + e_{u,y} = e_v * e_u. \tag{3}$$

However, it is impossible to execute this dynamic programming because there are infinitely many states. In the following, we propose a method to make the number of states finite and speed up the computation of this dynamic programming.

**Speeding up dynamic programming.** The most important insight for speeding up the computation is that $t_v$ and $e_v$ are convex piece-wise linear functions.

**Lemma 5.** $t_v$ and $e_v$ are convex and piece-wise linear functions with $O(|\mathcal{X}_v|)$ segments.

Intuitively, they are convex because nearby sinks are filled, and it costs more to transport extra mass as the amount of mass increases. To manage convex piece-wise linear functions efficiently, we represent each function by a sequence of slopes and lengths of segments, or equivalently, by the run-length representation of the convex conjugate function. Specifically, we represent a convex piecewise function $g$ as a tuple of $m(g) = \min_x g(x)$, $b(g) = \text{argmin}_x g(x) = \inf\{x \mid g(x) = m(g)\}$, and $B(g) = \{B(g)_i\}_{i=1,2,\ldots,n(g)}$, where $B(g)_i = (s(g)_i, l(g)_i)$ is a pair of slope $s(g)_i$ and length $l(g)_i$ of the $i$-th segment. It is known that min-sum convolution of convex functions is easy to compute in convex conjugate form.

**Lemma 6** (convex min-sum convolution [60]). *Let $f$ and $g$ be arbitrary convex piecewise functions. Then, $m(f * g) = m(f) + m(g)$, $b(f * g) = b(f) + b(g)$, and $B(f * g) = \text{sorted}(B(f) \| B(g))$, i.e., $B(f * g)$ is obtained by lining up elements of $B(f)$ and $B(g)$ in ascending order of slopes.*

In this form, for each leaf $v$, the initial state is $m(t_v) = 0$, $b(t_v) = \boldsymbol{b}_v - \boldsymbol{a}_v$, and $B(t_v) = \{(-\lambda_c(v), \infty), (\lambda_d(v), \infty)\}$ from Eq. (1). From Eq. (2), extending a subtree decreases the slopes by $w(v, p(v))$ where $x < 0$ and increases them by $w(v, p(v))$ where $x \geq 0$ and changes $m(e_v)$ and

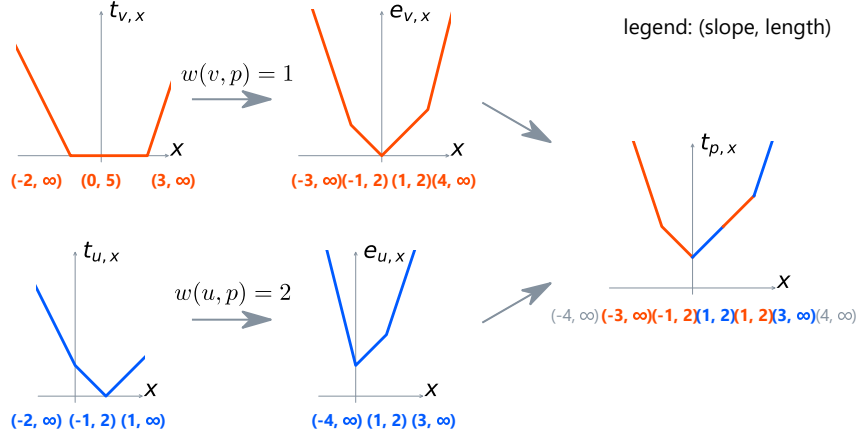

Figure 3: **Example of DP computation.** Our algorithm manages a piece-wise linear function by a set of the slopes and lengths of the segments.

$b(e_v)$ accordingly. From Eq. (3) and Lemma 6, merging two extended subtrees sorts $B(f_v)$ and $B(f_u)$ in increasing order of slopes, and $m(t_p)$ and $b(t_p)$ are the summations of the counterparts. Thanks to this re-formulation, the number of states is finite and the dynamic programming can be computed in $O(n^2 \log n)$ time because each node $v$ manipulates and sorts arrays of size $O(|\mathcal{X}_v|)$. However, this complexity is still unsatisfactory.

To speed up the computation, we use an efficient data structure. Specifically, we use a balanced binary tree, such as a red black tree, with additional information in each node [20, §14] to manage a set of segments in increasing order of slopes, or equivalently, in the increasing order of positions because the functions are convex. Eq. (2) can be computed in $O(\log m)$ time, where $m$ is the number of segments in a set, by lazy propagation. Eq. (2) can be computed in $O(\log m)$ time by inserting two segments into the balanced binary tree. The only obstacle is Eq. (3), where merging two sets may take $O(|\mathcal{T}_v| + |\mathcal{T}_u|)$ time. This problem can be solved by weighted-union heuristics [20]. Specifically, if the smaller set is always merged into the larger set, the total number of operations needed is $O(n \log n)$ in total. Because each insert operation requires $O(\log n)$ time, the total complexity is $O(n \log^2 n)$. Algorithm 1 describes the pseudo code of the algorithm.

**Theorem 7.** *Problem 4 can be solved exactly in $O(n \log^2 n)$ time.*

Because many unbalanced OT problems are special cases of GKR, they can also be computed in quasi-linear time.

**Corollary 8.** *The Kantorovich Rubinstein [41, 31], optimal partial transport [12, 27], Figalli [28, 35], and Lacombe [39] distances can be computed exactly in quasi-linear time on a tree metric.*

Besides, according to Theorem 3, tree GKR can approximate Euclidean GKR. Therefore, our algorithm can also *approximate* Euclidean GKR problems.

**Optimal Coupling.** Our algorithm can also reconstruct the optimal coupling $\pi^* \in \mathcal{U}^-(\boldsymbol{a}, \boldsymbol{b})$ efficiently by backtracking the DP table. This can be easily done by storing which segment is from which node and checking whether each segment is in the negative $x$ or positive $x$ in the root. Once the amount of mass creation and destruction at each node is computed, it is easy to compute the optimal coupling of the standard OT on a tree [5, 40]. This indicates that Flowtree [5] can be combined with the tree GKR to approximate the Euclidean GKR distance more accurately. We leave this direction as future work.

## 6   Experiments

We confirm the effectiveness of the proposed method through numerical experiments. We run the experiments on a Linux server with an Intel Xeon CPU E7-4830 @ 2.00 GHz and 1 TB RAM. We aim to answer the following questions. (Q1) How fast is the proposed method? (Q2) How accurately can tree GKR approximate Euclidean GKR? (Q3) Is the proposed method applicable to large-scale datasets? We also investigate the noise robustness of GKR, but we defer this to the appendices

**Algorithm 1:** GKR($\mathcal{T}, \boldsymbol{a}, \boldsymbol{b}, \lambda_d, \lambda_c, v$)

**1 Data:** Tree $\mathcal{T}$, Measures $\boldsymbol{a}, \boldsymbol{b}$, Cost functions $\lambda_d, \lambda_c$, and Node $v$.

**2 Result:** $t_v = (m(t_v), b(t_v), B(t_v))$

**3 begin**

**4**     $B(t_v) \leftarrow \{(-\lambda_c(v), \infty), (\lambda_d(v), \infty)\}; m(t_v) \leftarrow 0; b(t_v) \leftarrow \boldsymbol{b}_v - \boldsymbol{a}_v;$      // base case

**5**     **for** $c$: *child of* $v$ **do**

**6**        $t_c \leftarrow$ GKR($\mathcal{T}, \boldsymbol{a}, \boldsymbol{b}, \lambda_d, \lambda_c, c$)

**7**        Subtract $w(c, v)$ from the slopes of segments of $t_c$ where $x < 0$ and add $w(c, v)$ to the slopes of segments of $t_c$ where $x \geq 0$.      // Eq.(2)

**8**        **if** $|B(t_c)| > |B(t_v)|$ **then**

**9**           swap($B(t_c), B(t_v)$)      // weighted-union heuristics

**10**        $m(t_v) \leftarrow m(t_v) + m(t_c); b(t_v) \leftarrow b(t_v) + b(t_c)$

        **for** $s$: segments in $B(t_c)$ **do**

**11**           Insert $s$ into $B(t_v)$      // Eq.(3)

---

because the priority of this work is not discussing the usefulness of unbalanced OT, which has been extensively proved in existing works, but we aim at providing an efficient method for unbalanced OT.

**Speed Comparison:** We first measure the speed of the proposed method to show its efficiency. We use two baseline methods. The first one is the minimum cost flow-based algorithm, which takes as input the augmented graph as proposed by Guittet [31]. The second one is the Sinkhorn algorithm, where the cost matrix is the shortest distance matrix of the augmented graph. We use the network simplex algorithm in the Lemon graph library [25] as the implementation of the minimum cost flow algorithm and the C++ implementation of the Sinkhorn algorithm [56]. For each $n = 2^7, 2^8, \ldots, 2^{20}$, we generate 10 random trees with $n$ nodes. The amount of mass

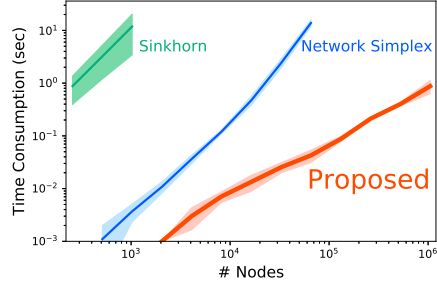

Figure 4: Speed comparison.

in each node is an integer drawn from an i.i.d. uniform distribution from 0 to $10^6$. The weight of each edge is also drawn from the same distribution. Figure 4 plots the time consumption of each method. The proposed method is several orders of magnitude faster than both baseline methods, and the difference is likely to increase as the data size increases. We also conduct the same experiments with a laptop computer with an Intel Core i5-7200U @ 2.50 GHz and 4 GB RAM. The proposed method processes a tree with $2^{20} (> 10^6)$ nodes in 0.70 second on this laptop.

**Approximation Accuracy:** We then measure the accuracy of the approximation of the quadtree GKR. We compute the GKR distances in the Euclidean space *exactly* using the network flow algorithm and compute them to the approximation by the quadtree GKR. We use Chicago Crime dataset[2], where each measure corresponds to a day and contains Dirac mass in the place where a crime occurred in that day. We approximate the GKR distance of measures endowed with the Euclidean distance by the GKR distance with the quadtree metric. We randomly sample 1000 pairs of days from January 2, 2015 to December 31, 2015 and compute the GKR distance $\text{GKR}_\text{euc}$ of these measures in the Euclidean space *exactly* with $\lambda_d = \lambda_c = \lambda \in [10^{-4}, 10.0]$. We then compute the GKR distance $\text{GKR}_\text{tree}$ of these measures using the quadtree of depth 15 without random translation. We linearly scale the quadtree metric because the scales of these distances are different. The scale is determined by 10 training data so that the relative error is minimized (see Appendix E). We compute the relative error $|\text{GKR}_\text{euc} - \text{GKR}_\text{tree}|/\text{GKR}_\text{euc}$ and the Spearman's rank correlation coefficient $\rho$ between $\text{GKR}_\text{euc}$ and $\text{GKR}_\text{tree}$ using the remaining 990 pairs of measures. Figure 5 shows that the correlation coefficient is larger than 0.9 for most $\lambda$, and the relative error is less than 0.1 for most cases. This indicates that

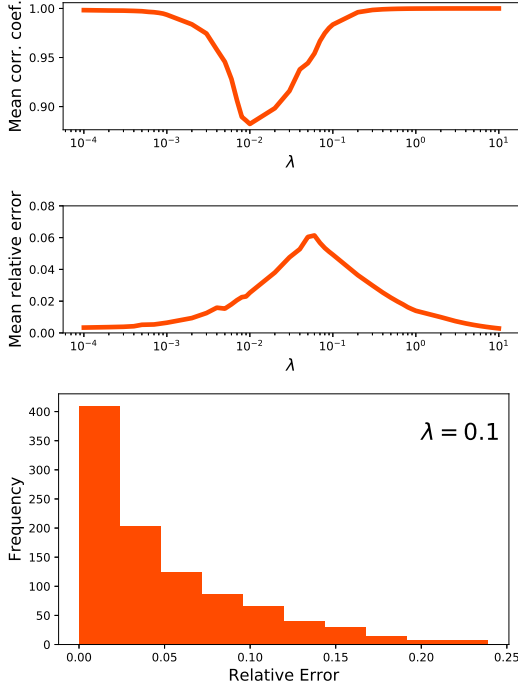

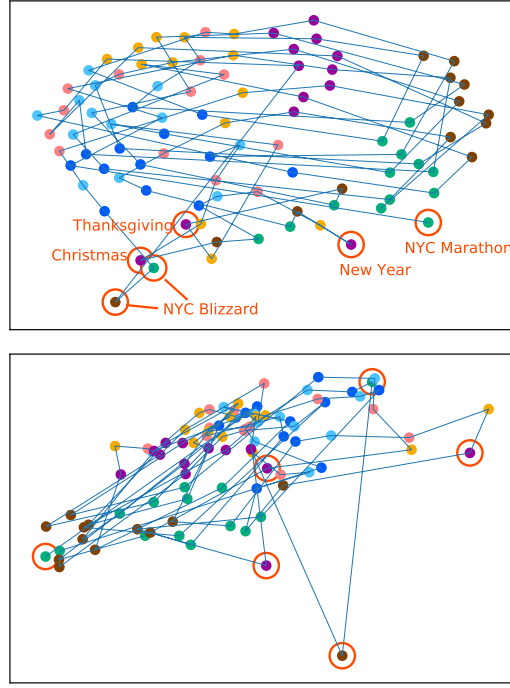

Figure 5: (Top) Spearman's $\rho$. (Mid, Bottom) Relative error.

Figure 6: MDS plots of (Top) tree GKR and (Bottom) tree sliced Wasserstein.

tree GKR can well approximate Euclidean GKR. Note that Theorem 3 guarantees approximation accuracy beyond two dimensions. The experiments in higher dimensions are reported in Appendix F

**Case Study (Large-Scale Dataset):** We apply the GKR distance to a large-scale dataset, namely, the NewYork taxi dataset[3] from November 1, 2015 to January 31, 2016. This dataset contains more than 66 million events, which is a scale never seen before in the unbalanced OT literature. The ground space is 2-dimensional space $\mathbb{R}^2$, and each mass represents a taxi pickup or dropoff event. We compute the distance between each pair of two days using the tree sliced Wasserstein (i.e., $\lambda_c = \lambda_d = \infty$) and GKR distance with $\lambda_c = \lambda_d = 0.001$ with quadtree. We normalize measures so that the total mass is equal to one for the tree sliced Wasserstein and use raw measures for GKR. Figure 6 plots the results of multidimensional scaling of both distances. Each dot represents a day and a color represents a day of the week. The GKR distance locates anomaly days, such as Christmas, new year, and severe blizzard days, to the bottom, weekends to the right, and weekdays to the left. Moreover, a clear periodicity can be seen in the GKR plot. The tree sliced Wasserstein also separates weekends and weekdays but does not locate anomaly days in similar positions. This is because GKR takes the number of events into account, whereas the standard OT does not due to normalization. The merits of both methods depend on the application. If one wants to know the probabilistic distribution of events, the standard OT is more appropriate, but if one wants to distinguish the amount of mass, the GKR distance is beneficial. GKR can balance this trade-off by setting parameters $\lambda_c$ and $\lambda_d$.

## 7    Conclusion

In this paper, we proposed the GKR distance, which encompasses many previous unbalanced OT problems. We showed that important special cases of the GKR distance on $L_p$ metrics cannot be computed in strongly subquadratic time under SETH. We then proposed a quasi-linear time algorithm to compute the GKR distance on tree metrics. Our algorithm can process more than one million masses in one second and can be applied to large-scale problems.

## Broader Impact

This work involves no ethical aspects. Because our algorithm reduces massive amount of computation, this work contributes to society by reducing power consumption and carbon footprint.

## Acknowledgments and Funding Disclosure

This work was supported by the JSPS KAKENHI GrantNumber 20H04243 and 20H04244.

## Footnotes

[2] https://data.cityofchicago.org/

[3] https://www1.nyc.gov/site/tlc/about/tlc-trip-record-data.page

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
