[Supplementary Material]

# A Proofs

*Proof of Theorem 2.* The Kantorovich Rubinstein (KR) distance is defined as follows.

$$\text{KR}(\mu, \nu) = \min_{\pi \in \mathcal{U}^-(\mu,\nu)} \int_{\mathcal{X} \times \mathcal{X}} c(x,y)d\pi(x,y) + \lambda \|\mu - \text{proj}_1 \pi\|_1 + \lambda \|\nu - \text{proj}_2 \pi\|_1$$

where $\lambda \in \mathbb{R}^+$ is a parameter. When $\lambda_c(x) = \lambda_d(x) = \lambda$ ($\forall x \in \mathcal{X}$), the GKR distance recovers the KR distance. The optimal partial transport distance is defined as follows.

$$\text{OPT}(\mu, \nu) = \min_{\pi \in \mathcal{U}_\kappa(\mu,\nu)} \int_{\mathcal{X} \times \mathcal{X}} c(x,y)d\pi(x,y),$$

where $\kappa \leq \min(\|\mu\|_1, \|\nu\|_1)$ is a parameter and $\mathcal{U}_\kappa(\mu, \nu) = \{\pi \in \mathcal{M}(\mathcal{X} \times \mathcal{X}) \mid \text{proj}_1 \pi \leq \mu, \text{proj}_2 \pi \leq \nu, \|\pi\|_1 = \kappa\}$ is the set of partial couplings. We prove the theorem by reduction from the following problem.

**Problem 9** (Bichromatic Hamming Close Pair (BHCP) problem [2]). **Input:** Two sets $\mathcal{A}, \mathcal{B} \subset \{0,1\}^d$ of $n$ binary vectors. **Output:** The closest pair distance $\min_{a \in \mathcal{A}, b \in \mathcal{B}} \|a - b\|_1$.

**Lemma 10** (Alman et al. [2]). *If there exists an algorithm that solves Problem 9 in $O(n^{2-\varepsilon})$ time for $d = \omega(\log n)$ and some $\varepsilon > 0$, SETH is false.*

**The KR distance:** We prove the contraposition of Theorem 2 for the KR distance. Suppose there exists an algorithm that computes the KR distance in $O(n^{2-\varepsilon})$ time for some $\varepsilon > 0$ on $d = \omega(\log n)$ dimensional $L_p$ metric space. We solve the BHCP problem using this algorithm. Let $\mathcal{A}, \mathcal{B} \subset \{0,1\}^d$ be any sets of $n$ binary vectors and $\delta = \min_{a \in \mathcal{A}, b \in \mathcal{B}} \|a - b\|_1$. Let $\mu = \sum_{a \in \mathcal{A}} \delta_a$ and $\nu = \sum_{b \in \mathcal{B}} \delta_b$, where $\delta_x$ is a Dirac mass at $x$. For $a \in \mathcal{A}, b \in \mathcal{B}$, we use $c(a, b) = \|a - b\|_p^p = \|a - b\|_1$ as a cost function. We prove that $\text{KR}(\mu, \nu) = 2\lambda n$ if $\lambda \leq \delta/2$ and $\text{KR}(\mu, \nu) < 2\lambda n$ otherwise. If $\lambda \leq \delta/2$,

$\text{KR}(\mu, \nu)$

$$= \min_{\pi \in \mathcal{U}^-(\mu,\nu)} \sum_{x \in \mathcal{A}} \sum_{y \in \mathcal{B}} c(x,y)\pi_{x,y} + \lambda \sum_{x \in \mathcal{A}} \left(\mu_x - \sum_{y \in \mathcal{B}} \pi_{x,y}\right) + \lambda \sum_{y \in \mathcal{B}} \left(\nu_y - \sum_{x \in \mathcal{A}} \pi_{x,y}\right)$$

$$\geq \min_{\pi \in \mathcal{U}^-(\mu,\nu)} \sum_{x \in \mathcal{A}} \sum_{y \in \mathcal{B}} 2\lambda\pi_{x,y} + \lambda \sum_{x \in \mathcal{A}} \left(\mu_x - \sum_{y \in \mathcal{B}} \pi_{x,y}\right) + \lambda \sum_{y \in \mathcal{B}} \left(\nu_y - \sum_{x \in \mathcal{A}} \pi_{x,y}\right)$$

$$= \min_{\pi \in \mathcal{U}^-(\mu,\nu)} 2\lambda \sum_{x \in \mathcal{A}} \sum_{y \in \mathcal{B}} 2\lambda\pi_{x,y} + \lambda \sum_{x \in \mathcal{A}} \mu_x - \lambda \sum_{x \in \mathcal{A}} \sum_{y \in \mathcal{B}} \lambda\pi_{x,y} + \lambda \sum_{y \in \mathcal{B}} \nu_y - \lambda \sum_{y \in \mathcal{B}} \sum_{x \in \mathcal{A}} \pi_{x,y}$$

$$= \lambda \sum_{x \in \mathcal{A}} \mu_x + \lambda \sum_{y \in \mathcal{B}} \nu_y = 2\lambda n.$$

Moreover, if $\lambda > \delta/2$, we take $a^* \in \mathcal{A}$ and $b^* \in \mathcal{B}$ such that $\|a^* - b^*\|_1 < 2\lambda$. Let $\pi^*$ be a subcoupling such that $\pi(a^*, b^*) = 1$ and $\pi(a, b) = 0$ if $a \neq a^*$ or $b \neq b^*$. Then,

$\text{KR}(\mu, \nu)$

$$= \min_{\pi \in \mathcal{U}^-(\mu,\nu)} \sum_{x \in \mathcal{A}} \sum_{y \in \mathcal{B}} c(x,y)\pi_{x,y} + \lambda \sum_{x \in \mathcal{A}} \left(\mu_x - \sum_{y \in \mathcal{B}} \pi_{x,y}\right) + \lambda \sum_{y \in \mathcal{B}} \left(\nu_y - \sum_{x \in \mathcal{A}} \pi_{x,y}\right)$$

$$\leq \sum_{x \in \mathcal{A}} \sum_{y \in \mathcal{B}} c(x,y)\pi_{x,y}^* + \lambda \sum_{x \in \mathcal{A}} \left(\mu_x - \sum_{y \in \mathcal{B}} \pi_{x,y}^*\right) + \lambda \sum_{y \in \mathcal{B}} \left(\nu_y - \sum_{x \in \mathcal{A}} \pi_{x,y}^*\right)$$

$$= \|a^* - b^*\|_1 \pi(a^*, b^*) + \lambda \sum_{x \in \mathcal{A}} \mu_x - \lambda \sum_{x \in \mathcal{A}} \sum_{y \in \mathcal{B}} \lambda\pi_{x,y}^* + \lambda \sum_{y \in \mathcal{B}} \nu_y - \lambda \sum_{y \in \mathcal{B}} \sum_{x \in \mathcal{A}} \pi_{x,y}^*$$

$$< 2\lambda + \lambda n - \lambda + \lambda n - \lambda = 2\lambda n.$$

Therefore, a binary search algorithm can determine $\delta$ by calling an algorithm for the Kantorovich Rubinstein distance in $O(\log n)$ time because $\delta$ is an integer between $0$ and $n$. This means that we can solve the BHCP problem in $O(n^{2-\varepsilon} \log n) \lesssim O(n^{2-\varepsilon/2})$ time. From Lemma 10, this indicates that SETH is false.

**Optimal partial transport:** We prove the contraposition of Theorem 2 for the optimal partial transport distance. Suppose there exists an algorithm that computes the optimal partial transport distance in $O(n^{2-\varepsilon})$ time for some $\varepsilon > 0$ in $d = \omega(\log n)$ dimensional $L_p$ metric space. We solve the BHCP problem using this algorithm. Let $\mathcal{A}, \mathcal{B} \subset \{0,1\}^d$ be any sets of $n$ binary vectors and $\delta = \min_{\boldsymbol{a} \in \mathcal{A}, \boldsymbol{b} \in \mathcal{B}} \|\boldsymbol{a} - \boldsymbol{b}\|_1$. Let $\mu = \sum_{\boldsymbol{a} \in \mathcal{A}} \delta_{\boldsymbol{a}}$ and $\nu = \sum_{\boldsymbol{b} \in \mathcal{B}} \delta_{\boldsymbol{b}}$, where $\delta_{\boldsymbol{x}}$ is a Dirac mass at $\boldsymbol{x}$. For $\boldsymbol{a} \in \mathcal{A}, \boldsymbol{b} \in \mathcal{B}$, we use $c(\boldsymbol{a}, \boldsymbol{b}) = \|\boldsymbol{a} - \boldsymbol{b}\|_p^p = \|\boldsymbol{a} - \boldsymbol{b}\|_1$ as a cost function. Then, if we set $\kappa = 1$, the optimal partial transport distance recovers the closest pair distance. Therefore, we can solve the BHCP problem in $O(n^{2-\varepsilon})$ time. From Lemma 10, this indicates that SETH is false. $\qquad\square$

*Proof of Theorem 3.*

**Lemma 11** ([33]). *Let $OT_{euc}$ be the OT distance with Euclidean cost $c_{euc}(x,y) = \|x - y\|_2$. Let $OT_{tree}$ be the OT distance with quadtree cost $c_{tree}(x,y) = d_{\mathcal{T}}(x,y)$, where $\mathcal{T}$ is a quadtree. There exists a constant $C_{OT}$ such that for any measures $\mu$ and $\nu$, $OT_{euc}(\mu, \nu) \leq C_{OT} \cdot OT_{tree}(\mu, \nu)$ holds. If we randomly translate measures when we construct a quadtree, there exists a constant $D_{OT}$ such that $\mathbb{E}_{\mathcal{T}}[OT_{tree}(\mu, \nu)] \leq D_{OT} \cdot OT_{euc}(\mu, \nu) \log \Delta$, where $\Delta$ is the spread.*

**Upper bound.** We first prove that $\mathrm{GKR}_{euc}(\mu, \nu) \leq C \cdot \mathrm{GKR}_{tree}(\mu, \nu)$. Let $C = \max(1, C_{OT})$ and $\pi_{tree}^*$ be the optimal coupling of $\mathrm{GKR}_{tree}(\mu, \nu)$.

$$C \cdot \mathrm{GKR}_{tree}(\mu, \nu)$$

$$= C \int_{\mathcal{X} \times \mathcal{X}} c_{tree} d\pi_{tree}^* + C \int_{\mathcal{X}} \lambda_d d(\mu - \mathrm{proj}_1 \pi_{tree}^*) + C \int_{\mathcal{X}} \lambda_c d(\nu - \mathrm{proj}_2 \pi_{tree}^*)$$

$$= C \cdot \mathrm{OT}_{tree}(\mathrm{proj}_1 \pi_{tree}^*, \mathrm{proj}_2 \pi_{tree}^*) + C \int_{\mathcal{X}} \lambda_d d(\mu - \mathrm{proj}_1 \pi_{tree}^*) + C \int_{\mathcal{X}} \lambda_c d(\nu - \mathrm{proj}_2 \pi_{tree}^*)$$

$$\geq C_{OT} \cdot \mathrm{OT}_{tree}(\mathrm{proj}_1 \pi_{tree}^*, \mathrm{proj}_2 \pi_{tree}^*) + \int_{\mathcal{X}} \lambda_d d(\mu - \mathrm{proj}_1 \pi_{tree}^*) + \int_{\mathcal{X}} \lambda_c d(\nu - \mathrm{proj}_2 \pi_{tree}^*)$$

$$\geq \mathrm{OT}_{euc}(\mathrm{proj}_1 \pi_{tree}^*, \mathrm{proj}_2 \pi_{tree}^*) + \int_{\mathcal{X}} \lambda_d d(\mu - \mathrm{proj}_1 \pi_{tree}^*) + \int_{\mathcal{X}} \lambda_c d(\nu - \mathrm{proj}_2 \pi_{tree}^*)$$

$$\geq \mathrm{GKR}_{euc}(\mu, \nu)$$

**Lower bound.** We then prove that $\mathbb{E}_{\mathcal{T}}[\mathrm{GKR}_{tree}(\mu, \nu)] \leq D \cdot \mathrm{GKR}_{euc}(\mu, \nu) \log \Delta$. Let $D = \max(\frac{1}{\log \Delta}, D_{OT})$ and $\pi_{euc}^*$ be the optimal coupling of $\mathrm{GKR}_{euc}(\mu, \nu)$.

$$D \cdot \mathrm{GKR}_{euc}(\mu, \nu) \log \Delta$$

$$= D \log \Delta \int_{\mathcal{X} \times \mathcal{X}} c_{euc} d\pi_{euc}^* + D \log \Delta \int_{\mathcal{X}} \lambda_d d(\mu - \mathrm{proj}_1 \pi_{euc}^*) + D \log \Delta \int_{\mathcal{X}} \lambda_c d(\nu - \mathrm{proj}_2 \pi_{euc}^*)$$

$$= D \log \Delta \cdot \mathrm{OT}_{euc}(\mathrm{proj}_1 \pi_{euc}^*, \mathrm{proj}_2 \pi_{euc}^*) + D \log \Delta \int_{\mathcal{X}} \lambda_d d(\mu - \mathrm{proj}_1 \pi_{euc}^*) + D \log \Delta \int_{\mathcal{X}} \lambda_c d(\nu - \mathrm{proj}_2 \pi_{euc}^*)$$

$$\geq D_{OT} \log \Delta \cdot \mathrm{OT}_{euc}(\mathrm{proj}_1 \pi_{euc}^*, \mathrm{proj}_2 \pi_{euc}^*) + \int_{\mathcal{X}} \lambda_d d(\mu - \mathrm{proj}_1 \pi_{euc}^*) + \int_{\mathcal{X}} \lambda_c d(\nu - \mathrm{proj}_2 \pi_{euc}^*)$$

$$\geq \mathbb{E}_{\mathcal{T}}[\mathrm{OT}_{tree}(\mathrm{proj}_1 \pi_{tree}^*, \mathrm{proj}_2 \pi_{tree}^*)] + \int_{\mathcal{X}} \lambda_d d(\mu - \mathrm{proj}_1 \pi_{euc}^*) + \int_{\mathcal{X}} \lambda_c d(\nu - \mathrm{proj}_2 \pi_{euc}^*)$$

$$= \mathbb{E}_{\mathcal{T}}[\mathrm{OT}_{tree}(\mathrm{proj}_1 \pi_{tree}^*, \mathrm{proj}_2 \pi_{tree}^*) + \int_{\mathcal{X}} \lambda_d d(\mu - \mathrm{proj}_1 \pi_{euc}^*) + \int_{\mathcal{X}} \lambda_c d(\nu - \mathrm{proj}_2 \pi_{euc}^*)]$$

$$\geq \mathbb{E}_{\mathcal{T}}[\mathrm{GKR}_{tree}(\mu, \nu)]$$

$$\qquad\square$$

*Proof of Lemma 5.* In a leaf node $v$, $t_v$ is convex from Eq.(1). If $t_x$ is convex, $e_x$ is convex from Eq. (2) because both $|x| \cdot w(v, p(v))$ and $t_x$ are convex. If $e_v$ and $e_u$ are convex, $t_{p,x}$ is convex from Eq. (3). Therefore, $t_v$ and $e_v$ are convex by induction. Next, we prove that $t_v$ and $e_v$ are

piece-wise constant with at most $3|\mathcal{X}_v|$ segments. In a leaf node $v$, $|B(t_v)| = 2 \leq 3$ from Eq. (1) and $|B(e_v)| \leq |B(t_v)| + 1 = 3 \leq 3$ from Eq. (2). In an internal node $p$ with children $v$ and $u$, $|B(t_p)| = |B(t_v)| + |B(t_u)| \leq 3(|\mathcal{X}_p| - 1)$ from Eq. (1) and the inductive hypothesis and $|B(e_p)| \leq |B(t_p)| + 1 \leq 3|\mathcal{X}_p| - 2$ from Eq. (2). Therefore, $t_v$ and $e_v$ are piece-wise constant with at most $3|\mathcal{X}_v|$ segments. $\qquad \square$

*Proof of Theorem 7.* In each node, computing Eq. (1) (i.e., Line 4 in Algorithm 1) requires $O(1)$ time. Computing Eq. (2) (i.e., Line 7 in Algorithm 1) requires $O(\log |\mathcal{X}_x|) \lesssim O(\log n)$ time because adding a constant to elements of a range of a balanced binary tree requires logarithmic time, and the number of elements in the balanced binary tree is $O(|\mathcal{X}_x|)$ from Lemma 5. Therefore, computing Eq. (2) require $O(n \log n)$ time in total. Due to the weighted-union heuristics, there are $O(n \log n)$ insertion operations to compute Eq. (3) (i.e., Line 11 in Algorithm 1) in total. Because an insertion operation of a balanced binary tree requires logarithmic time, computing Eq. (3) requires $O(n \log^2 n)$ time in total. Therefore, the total time complexity is $O(n \log^2 n)$. $\qquad \square$

## B  Triangle Inequality

We prove that the triangle inequality holds if the two conditions mentioned in the main text hold. Intuitively, it is cheaper to transport/create/destruct mass directly than to transport them to intermediate places or to create/destruct at intermediate places. We provide a proof for the discrete case. The continuous case can be proved similarly.

**Theorem 12.** $GKR(\mu, \eta) \leq GKR(\mu, \nu) + GKR(\nu, \eta)$ *holds for any* $\mu = \sum_{x \in \mathcal{X}} a_x \delta_x$, $\nu = \sum_{x \in \mathcal{X}} b_x \delta_x$, $\eta = \sum_{x \in \mathcal{X}} c_x \delta_x$ *if (1) cost* $d$ *is a metric and (2)* $\lambda_d(x) \leq c(x, y) \lambda_d(y)$ *and* $\lambda_c(y) \leq \lambda_c(x) + c(x, y)$ *hold for any* $x, y \in \mathcal{X}$.

*Proof.* Let $P$ and $Q$ be the optimal transportation matrix for $GKR(\mu, \nu)$ and $GKR(\nu, \eta)$. Thus,

$$a_x \geq \sum_{y \in \mathcal{X}} P_{x,y} = \text{proj}_1 P_x \quad (\forall x \in \mathcal{X}),$$

$$b_y \geq \sum_{x \in \mathcal{X}} P_{x,y} = \text{proj}_2 P_y \quad (\forall y \in \mathcal{X}),$$

$$GKR(\mu, \nu) = \sum_{x,y \in \mathcal{X}} P_{x,y} d(x, y) + \sum_{x \in \mathcal{X}} \lambda_d(x) \left( a_x - \sum_{y \in \mathcal{X}} P_{x,y} \right) + \sum_{y \in \mathcal{X}} \lambda_c(y) \left( b_y - \sum_{x \in \mathcal{X}} P_{x,y} \right),$$

$$b_y \geq \sum_{z \in \mathcal{X}} Q_{y,z} = \text{proj}_1 Q_y \quad (\forall y \in \mathcal{X}),$$

$$c_z \geq \sum_{y \in \mathcal{X}} Q_{y,z} = \text{proj}_2 Q_z \quad (\forall z \in \mathcal{X}),$$

$$GKR(\nu, \eta) = \sum_{y,z \in \mathcal{X}} Q_{y,z} d(y, z) + \sum_{y \in \mathcal{X}} \lambda_d(y) \left( b_y - \sum_{z \in \mathcal{X}} Q_{y,z} \right) + \sum_{z \in \mathcal{X}} \lambda_c(z) \left( c_z - \sum_{y \in \mathcal{X}} Q_{y,z} \right).$$

Let $\mathcal{P} = \{ y \in \mathcal{X} \mid \text{proj}_2 P_y > \text{proj}_1 Q_y > 0 \}$ and $\mathcal{Q} = \{ y \in \mathcal{X} \mid \text{proj}_1 Q_y > \text{proj}_2 P_y > 0 \}$, and

$$R_{xz} = \sum_{y \in \mathcal{P} \cup \mathcal{Q}} \frac{P_{x,y} Q_{y,z}}{\max(\text{proj}_2 P_y, \text{proj}_1 Q_y)}.$$

Then,

$$GKR(\mu, \eta) \leq \sum_{x,z \in \mathcal{X}} R_{x,z} d(x, z) + \sum_{x \in \mathcal{X}} \lambda_d(x) \left( a_x - \sum_{z \in \mathcal{X}} R_{x,z} \right) + \sum_{z \in \mathcal{X}} \lambda_c(z) \left( c_z - \sum_{x \in \mathcal{X}} R_{x,z} \right)$$

$$= \sum_{x,z \in \mathcal{X}} \sum_{y \in \mathcal{P}} \frac{P_{x,y} Q_{y,z}}{\text{proj}_2 P_y} d(x, z) + \sum_{x,z \in \mathcal{X}} \sum_{y \in \mathcal{Q}} \frac{P_{x,y} Q_{y,z}}{\text{proj}_1 Q_y} d(x, z)$$

$$+ \sum_{x \in \mathcal{X}} \lambda_d(x) \left( a_x - \sum_{z \in \mathcal{X}} \left( \sum_{y \in \mathcal{P}} \frac{P_{x,y} Q_{y,z}}{\text{proj}_2 P_y} + \sum_{y \in \mathcal{Q}} \frac{P_{x,y} Q_{y,z}}{\text{proj}_1 Q_y} \right) \right)$$

$$+ \sum_{z \in \mathcal{X}} \lambda_c(z) \left( c_z - \sum_{x \in \mathcal{X}} \left( \sum_{y \in \mathcal{P}} \frac{P_{x,y} Q_{y,z}}{\text{proj}_2 P_y} + \sum_{y \in \mathcal{Q}} \frac{P_{x,y} Q_{y,z}}{\text{proj}_1 Q_y} \right) \right)$$

$$\leq \sum_{x,z \in \mathcal{X}} \sum_{y \in \mathcal{P}} \frac{P_{x,y} Q_{y,z}}{\text{proj}_2 P_y} (d(x,y) + d(y,z)) + \sum_{x,z \in \mathcal{X}} \sum_{y \in \mathcal{Q}} \frac{P_{x,y} Q_{y,z}}{\text{proj}_1 Q_y} (d(x,y) + d(y,z))$$

$$+ \sum_{x \in \mathcal{X}} \lambda_d(x) \left( a_x - \sum_{z \in \mathcal{X}} \left( \sum_{y \in \mathcal{P}} \frac{P_{x,y} Q_{y,z}}{\text{proj}_2 P_y} + \sum_{y \in \mathcal{Q}} \frac{P_{x,y} Q_{y,z}}{\text{proj}_1 Q_y} \right) \right)$$

$$+ \sum_{z \in \mathcal{X}} \lambda_c(z) \left( c_z - \sum_{x \in \mathcal{X}} \left( \sum_{y \in \mathcal{P}} \frac{P_{x,y} Q_{y,z}}{\text{proj}_2 P_y} + \sum_{y \in \mathcal{Q}} \frac{P_{x,y} Q_{y,z}}{\text{proj}_1 Q_y} \right) \right)$$

$$= \sum_{x \in \mathcal{X}} \sum_{y \in \mathcal{P}} \frac{\text{proj}_1 Q_y}{\text{proj}_2 P_y} P_{x,y} d(x,y) + \sum_{z \in \mathcal{X}} \sum_{y \in \mathcal{P}} Q_{y,z} d(y,z)$$

$$+ \sum_{x \in \mathcal{X}} \sum_{y \in \mathcal{Q}} P_{x,y} d(x,y) + \sum_{z \in \mathcal{X}} \sum_{y \in \mathcal{Q}} \frac{\text{proj}_1 P_y}{\text{proj}_1 Q_y} Q_{y,z} d(y,z)$$

$$+ \sum_{x \in \mathcal{X}} \lambda_d(x) \left( a_x - \sum_{y \in \mathcal{P}} \frac{\text{proj}_1 Q_y}{\text{proj}_2 P_y} P_{x,y} - \sum_{y \in \mathcal{Q}} P_{x,y} \right)$$

$$+ \sum_{z \in \mathcal{X}} \lambda_c(z) \left( c_z - \sum_{y \in \mathcal{P}} Q_{y,z} - \sum_{y \in \mathcal{Q}} \frac{\text{proj}_2 P_y}{\text{proj}_1 Q_y} Q_{y,z} \right)$$

$$= \sum_{x \in \mathcal{X}} \sum_{y \in \mathcal{P}} \frac{\text{proj}_1 Q_y}{\text{proj}_2 P_y} P_{x,y} d(x,y) + \sum_{z \in \mathcal{X}} \sum_{y \in \mathcal{P}} Q_{y,z} d(y,z)$$

$$+ \sum_{x \in \mathcal{X}} \sum_{y \in \mathcal{Q}} P_{x,y} d(x,y) + \sum_{z \in \mathcal{X}} \sum_{y \in \mathcal{Q}} \frac{\text{proj}_1 P_y}{\text{proj}_1 Q_y} Q_{y,z} d(y,z)$$

$$+ \sum_{x \in \mathcal{X}} \lambda_d(x) \left( a_x - \sum_{y \in \mathcal{X}} P_{x,y} \right) + \sum_{z \in \mathcal{X}} \lambda_c(z) \left( c_z - \sum_{y \in \mathcal{X}} Q_{y,z} \right)$$

$$+ \sum_{x \in \mathcal{X}} \sum_{y \in \mathcal{P}} \lambda_d(x) \left( 1 - \frac{\text{proj}_1 Q_y}{\text{proj}_2 P_y} \right) P_{x,y} + \sum_{z \in \mathcal{X}} \sum_{y \in \mathcal{Q}} \lambda_c(z) \left( 1 - \frac{\text{proj}_2 P_y}{\text{proj}_1 Q_y} \right) Q_{y,z}$$

$$\leq \sum_{x \in \mathcal{X}} \sum_{y \in \mathcal{P}} \frac{\text{proj}_1 Q_y}{\text{proj}_2 P_y} P_{x,y} d(x,y) + \sum_{z \in \mathcal{X}} \sum_{y \in \mathcal{P}} Q_{y,z} d(y,z)$$

$$+ \sum_{x \in \mathcal{X}} \sum_{y \in \mathcal{Q}} P_{x,y} d(x,y) + \sum_{z \in \mathcal{X}} \sum_{y \in \mathcal{Q}} \frac{\text{proj}_1 P_y}{\text{proj}_1 Q_y} Q_{y,z} d(y,z)$$

$$+ \sum_{x \in \mathcal{X}} \lambda_d(x) \left( a_x - \sum_{y \in \mathcal{X}} P_{x,y} \right) + \sum_{z \in \mathcal{X}} \lambda_c(z) \left( c_z - \sum_{y \in \mathcal{X}} Q_{y,z} \right)$$

$$+ \sum_{x \in \mathcal{X}} \sum_{y \in \mathcal{P}} (d(x,y) + \lambda_d(y)) \left( 1 - \frac{\text{proj}_1 Q_y}{\text{proj}_2 P_y} \right) P_{x,y}$$

$$+ \sum_{z \in \mathcal{X}} \sum_{y \in \mathcal{Q}} (\lambda_c(y) + d(y,z)) \left( 1 - \frac{\text{proj}_2 P_y}{\text{proj}_1 Q_y} \right) Q_{y,z}$$

$$= \sum_{x \in \mathcal{X}} \sum_{y \in \mathcal{P}} P_{x,y} d(x,y) + \sum_{z \in \mathcal{X}} \sum_{y \in \mathcal{P}} Q_{y,z} d(y,z)$$

$$+ \sum_{x \in \mathcal{X}} \sum_{y \in \mathcal{Q}} P_{x,y} d(x,y) + \sum_{z \in \mathcal{X}} \sum_{y \in \mathcal{Q}} Q_{y,z} d(y,z)$$

$$+ \sum_{x \in \mathcal{X}} \lambda_d(x) \left( a_x - \sum_{y \in \mathcal{X}} P_{x,y} \right) + \sum_{z \in \mathcal{X}} \lambda_c(z) \left( c_z - \sum_{y \in \mathcal{X}} Q_{y,z} \right)$$

$$+ \sum_{x \in \mathcal{X}} \sum_{y \in \mathcal{P}} \lambda_d(y) \left( 1 - \frac{\text{proj}_1 Q_y}{\text{proj}_2 P_y} \right) P_{x,y}$$

$$+ \sum_{z \in \mathcal{X}} \sum_{y \in \mathcal{Q}} \lambda_c(y) \left( 1 - \frac{\text{proj}_2 P_y}{\text{proj}_1 Q_y} \right) Q_{y,z}$$

$$= \sum_{x \in \mathcal{X}} \sum_{y \in \mathcal{X}} P_{x,y} d(x,y) + \sum_{z \in \mathcal{X}} \sum_{y \in \mathcal{X}} Q_{y,z} d(y,z)$$

$$+ \sum_{x \in \mathcal{X}} \lambda_d(x) \left( a_x - \sum_{y \in \mathcal{X}} P_{x,y} \right) + \sum_{z \in \mathcal{X}} \lambda_c(z) \left( c_z - \sum_{y \in \mathcal{X}} Q_{y,z} \right)$$

$$+ \sum_{y \in \mathcal{P}} \lambda_d(y)(\text{proj}_2 P_y - \text{proj}_1 Q_y) + \sum_{y \in \mathcal{Q}} \lambda_c(y)(\text{proj}_1 Q_y - \text{proj}_2 P_y)$$

$$\leq \sum_{x \in \mathcal{X}} \sum_{y \in \mathcal{X}} P_{x,y} d(x,y) + \sum_{z \in \mathcal{X}} \sum_{y \in \mathcal{X}} Q_{y,z} d(y,z)$$

$$+ \sum_{x \in \mathcal{X}} \lambda_d(x) \left( a_x - \sum_{y \in \mathcal{X}} P_{x,y} \right) + \sum_{z \in \mathcal{X}} \lambda_c(z) \left( c_z - \sum_{y \in \mathcal{X}} Q_{y,z} \right)$$

$$+ \sum_{y \in \mathcal{X}} \lambda_d(y)(b_y - \text{proj}_1 Q_y) + \sum_{y \in \mathcal{X}} \lambda_c(y)(b_y - \text{proj}_2 P_y)$$

$$= \text{GKR}(\mu, \nu) + \text{GKR}(\nu, \eta).$$

$\square$

## C   Identity of Indiscernibles

We prove that $\text{GKR}(\mu, \nu) = 0$ iff $\mu = \nu$ under mild conditions in the discrete case.

**Theorem 13.** *$GKR(\mu, \nu) = 0$ iff $\mu = \nu$ holds for any $\mu = \sum_{x \in \mathcal{X}} a_x \delta_x$, $\nu = \sum_{x \in \mathcal{X}} b_x \delta_x$ if (1) $c(x,x) = 0$ for all $x \in \mathcal{X}$ and $c(x,y) > 0$ if $x \neq y$ and (2) $\lambda_c(x) > 0$ and $\lambda_d(x) > 0$ for all $x \in \mathcal{X}$.*

*Proof.* Suppose $\mu = \nu$. Let $P_{xy} = 0$ if $x \neq y$ and $P_{xx} = a_x(= b_x)$. Then, $P$ is a valid transportation matrix and

$$\sum_{x,y \in \mathcal{X}} P_{xy} c(x,y) + \sum_{x \in \mathcal{X}} \lambda_d(x) \left( a_x - \sum_{y \in \mathcal{X}} P_{x,y} \right) + \sum_{y \in \mathcal{X}} \lambda_c(y) \left( b_y - \sum_{x \in \mathcal{X}} P_{x,y} \right)$$

$$= \sum_{x \in \mathcal{X}} a_x c(x,x) + \sum_{x \in \mathcal{X}} \lambda_d(x)(a_x - a_x) + \sum_{x \in \mathcal{X}} \lambda_c(x)(b_x - b_x)$$

$$= 0.$$

Suppose $\text{GKR}(\mu, \nu) = 0$ and let $P$ be the optimal transportation matrix. Then,

$$a_x \geq \sum_{y \in \mathcal{X}} P_{x,y} = \text{proj}_1 P_x \quad (\forall x \in \mathcal{X}),$$

$$b_y \geq \sum_{x \in \mathcal{X}} P_{x,y} = \text{proj}_2 P_y \quad (\forall y \in \mathcal{X}),$$

$$\text{GKR}(\mu,\nu) = \sum_{x,y\in\mathcal{X}} P_{x,y}c(x,y) + \sum_{x\in\mathcal{X}}\lambda_d(x)\left(a_x - \sum_{y\in\mathcal{X}}P_{x,y}\right) + \sum_{y\in\mathcal{X}}\lambda_c(y)\left(b_y - \sum_{x\in\mathcal{X}}P_{x,y}\right) = 0.$$

Each term is zero because each term must be non-negative. In particular,

$$a_x = \sum_{y\in\mathcal{X}} P_{x,y}$$

$$b_y = \sum_{x\in\mathcal{X}} P_{x,y},$$

because $\lambda_d(x) > 0$ and $\lambda_c(x) > 0$ for all $x \in \mathcal{X}$. Because $c(x,y) > 0$ if $x \neq y$, $P_{x,y} = 0$ if $x \neq y$. Therefore,

$$a_x = \sum_{y\in\mathcal{X}} P_{x,y} = P_{x,x}$$

$$b_y = \sum_{x\in\mathcal{X}} P_{x,y} = P_{y,y},$$

which means that $a_x = b_x$ for all $x \in \mathcal{X}$ and that $\mu = \nu$. $\qquad\square$

## D Preprocessing for Analysis

We discuss the validity of the assumption that the input is a binary tree with mass only in leaf nodes. First, we attach dummy nodes with no mass to nodes so that all internal nodes have at least two children. For each internal node $v$, we create a child $v'$ with the same mass as $v$, connect $v$ and $v'$ by an edge with weight 0, and set the mass of $v$ as 0. Then, for each internal node with more than two children, we create a new child $v'$, connect $v$ and $v'$ by an edge with weight 0, and change the parent of two arbitrary children of $v$ to $v'$ recursively. Obviously, this transformation blows up the input size only linearly and does not change the GKR distance. Therefore, we can make the assumptions without loss of generality.

## E Scaling Quadtree

In the approximation error experiments, we scale the edge length of the quadtree using a training dataset so that the relative error $|\text{GKR}_{\text{euc}} - \text{GKR}_{\text{tree}}|/\text{GKR}_{\text{euc}}$ is minimized. Specifically, we search the scale parameter $s$ such that GKR with cost $c(x,y) = s \cdot d_{\mathcal{T}}(x,y)$ minimizes the relative error. We use the ternary search to determine the scale parameter. Although the relative error is not necessarily unimodal, we found this was a good heuristic to determine the scale parameter efficiently.

We also conduct experiments without any training dataset. We determine the scale parameter by simple heuristics instead of the ternary search. Specifically, we sample some pairs $(x_1, y_1), \ldots, (x_K, y_K)$ of nodes in the quadtree and use the average ratio $s = \frac{1}{K}\sum_{i=1}^{K}\frac{d_{\text{euc}}(x,y)}{d_{\mathcal{T}}(x,y)}$ of the Euclidean distance to the tree distance as the scale parameter. Note that this value is independent of the parameters $\lambda$ of the GKR distance, while the ternary search is dependent. In the Chicago crime dataset, the ratio is $s \approx 0.18$. Figure 7 shows the Spearman's rank correlation coefficient and the relative error of the same set of 990 pairs of measures as in the main experiment. This shows that the relative error is worse than that in Figure 5 because the scales of two distances are different, but the Spearman's rank correlation coefficient is comparable to that in Figure 5. When one classifies or visualizes measures, the relative order is important. The high rank correlation indicates that this simple heuristic is beneficial when no training data are available.

## F Approximation Accuracy in high dimensions

The quadtree is *NOT* restricted to two dimensions [33] and can be constructed efficiently even in high dimensional spaces [5]. However, the quadtree may degrade its empirical performance in high dimensional cases [40]. In that case, clustering-based trees can be used [40]. We conduct additional experiments in high dimensional cases. First, we compute GKR for the Chicago Crime dataset with

Figure 7: (Top) Spearman's $\rho$ and (Bottom) Relative error with the scale parameter determined without training data but with a simple heuristic.

Figure 8: Accuracy in high dimensional cases.

the additional time axis, using the *quadtree*. Each mass represents a crime in the 3-dimensional (longitude, latitude, time) space. We normalize each dimension so that each dimension has the same scale. Next, we compute GKR for the (unbalanced) Word Mover's Distance of the Twitter dataset [40] using *clustering-based trees*. Each measure represents a sentence, and each mass represents a word embedded in a 300-dimensional space computed by a pre-trained language model. We compare our algorithm with the ground-truth GKR distance in the Euclidean space, as we did in Section 6. Figure 8 shows that our algorithm can approximate high dimensional GKR accurately.

## G    Noise Robustness Experiments

We confirm the unbalanced OT distance is robust to noise using shape comparison experiments. We use cluttered MNIST [44] to this end, where the patch size is 4 with no translation operation. Figure 9 illustrates the dataset. For each $k = 0, 1, \ldots, 16$, we generate 10 shapes with $k$ clutters for each class. The ground space is a 2-dimensional lattice $\{0, 1, \ldots, 27\} \times \{0, 1, \ldots, 27\}$, and the amount of mass

in each point corresponds to the normalized brightness of the pixel. We use two baseline methods: the Sinkhorn algorithm and the tree sliced Wasserstein [40]. The tree sliced Wasserstein corresponds to GKR with $\lambda_c = \lambda_d = \infty$. The Sinkhorn algorithm uses the Euclidean distance between two masses as the cost matrix. We use quadtree for the tree sliced Wasserstein and GKR. We set $\lambda_c = \lambda_d = 8$ for the GKR distance. We classify each digit by 1-NN using each distance. Figure 10 plots the accuracy of each distance. The accuracies of all distances are comparable for $k = 0$ (i.e., no noise), but the GKR distance outperforms the other two methods for noisy shapes. This indicates that the GKR distance is robust to noise compared to the standard optimal transport distance.

**Comparison with Generalized Sinkhorn.** We also compared our algorithm with the generalized Sinkhorn algorithm [18] using the same setting. We compute *Euclidean* UOT (i.e., without tree approximation) using the generalized Sinkhorn algorithm and carry out 1-NN classification. We use the KL divergence as the regularizer and set the hyperparameters to $\varepsilon = 0.01$ and $\lambda = 0.01$. Figure 11 shows that the generalized Sinkhorn is also robust to noise compared to standard optimal transport distances. Since the generalized Sinkhorn requires at least $O(n^2)$ time, its applicability is limited to thousand-scale datasets. In contrast, our algorithm is applicable to million-scale datasets keeping its performance.

## H    Document Classification Experiments

To show further use-cases of GKR, we conducted document classification, following [38, 40]. We use the Twitter dataset [40]. Each mass represents a word, and each measure represents a document (i.e., a tweet). We use word2vec embedding trained on the Google News corpus for the word embedding. Thus, the ground space is a 300-dimensional Euclidean space. We compute the GKR distance between pairs of documents exactly (without tree approximation), and carry out 1-NN classification. Note that when $\lambda \to \infty$, GKR does not create nor delete mass due to high creation and destruction costs, thus GKR is reduced to standard OT (i.e., word mover's distance). We measure the accuracy using documents with more than three words in our evaluation. Figure 12 reports the performances of our algorithm and the word mover's distance. This result indicates that GKR performs well in document classification. This is intuitively because GKR can ignore noisy words thanks to the mass creation and destruction mechanism. We hypothesize that this tendency can be extended to other OT applications. We will explore more applications in future work.

Figure 9: Examples of the cluttered MNIST dataset.

Figure 10: 1-NN classification accuracy for the cluttered MNIST dataset.

Figure 11: Comparison with the generalized Sinkhorn algorithm using the cluttered MNIST dataset.

Table 1: 1-NN classification accuracy for the cluttered MNIST dataset.

| k | 0 | 1 | 2 | 3 | 4 | 5 | 6 | 7 | 8 | 9 | 10 | 11 | 12 | 13 | 14 | 15 | 16 |
|---|---|---|---|---|---|---|---|---|---|---|----|----|----|----|----|----|----|
| Tree GKR | **0.84** | **0.85** | **0.81** | **0.77** | **0.75** | **0.78** | **0.73** | **0.7** | **0.67** | **0.58** | **0.59** | **0.53** | **0.54** | **0.52** | **0.55** | **0.42** | **0.39** |
| Tree-sliced Wasserstein | 0.78 | 0.69 | 0.71 | 0.63 | 0.61 | 0.49 | 0.46 | 0.52 | 0.55 | 0.44 | 0.43 | 0.48 | 0.32 | 0.23 | 0.45 | 0.2 | 0.22 |
| Sinkhorn | **0.83** | 0.74 | 0.72 | 0.64 | 0.56 | 0.47 | 0.36 | 0.41 | 0.46 | 0.35 | 0.37 | 0.3 | 0.32 | 0.18 | 0.29 | 0.17 | 0.17 |
| generalized Sinkhorn | **0.83** | 0.76 | 0.75 | **0.78** | 0.59 | 0.63 | 0.62 | 0.63 | 0.57 | 0.47 | **0.56** | 0.41 | 0.39 | 0.40 | 0.39 | 0.36 | **0.37** |

Figure 12: 1-NN classification accuracy for the twitter dataset.