[Reviews · NeurIPS 2020]

Review 1

Summary and Contributions: The paper introduces a generalized formulation of unbalanced OT distances, called the generalized Kantorovich Rubinstein (GKR) distance. Several previous work become the variants of the GKR distance. The authors first prove the negative result that the KR distance and the partial OT distance can not be solved under O(n^2). Then, the authors propose to use the GKR distance on quad trees that are derived from the Euclidean space to approximate the GKR distance on the original Euclidean space as the ground metric. The approximation is proved to be upper- and lower-bounded. Moreover, the authors propose a fast algorithm to compute GKR, based on accelerated dynamic programming and optimized tree structures. The algorithm is proved to solve for the GKR distance on tree metrics in O(n(logn)^2). Some experiments wrap up the paper testing its complexity, approximation accuracy, and scalability.

Strengths: -- soundness of the claims: I haven't found any error in methodology and proofs. -- significance and novelty of the contribution: The paper introduces a fast OT algorithm that breaks the O(n^2) barrier. This is a significant algorithmic breakthrough. -- relevance to the NeurIPS community The paper addresses an important open problem of unbalanced OT that has numerous applications in the machine learning community.

Weaknesses: I included in here all my other comments and questions, some of which are not necessarily weaknesses. L 131: "the GKR distance does not necessarily transport all the mass but pays penalties for mass creation and destruction" The two hyper-parameters might be troublesome for practical use, especially when the authors do not offer any insights or cross-validation on that L 135: "... the GKR distance includes many popular variants of the OT distance as special cases" I suggest the authors include the mathematical expressions of the connections between GKR and other existing (unbalanced) OT distances, at least in the supplementary. -- Page 1, the noise toy experiment Line 20: The noise example might not best illustrate the advantage of unbalanced OT because most noises can be assumed to have a zero mean (after some preprocessing) and thus may have a very little impact on the normalization. Comparing measures that originally have different totals may better serve the purpose, such as two datasets with different number of Dirac samples or the total measure changes as the data evolve in a dynamic process. L 585: The baseline is missing an existing unbalanced OT solver. -- Page 7, Chicago Crime Specifying the ground metric as R^2 helps readers understand the experiment if they have no background of the dataset. What is the rationale behind a distance between measures in different days? Intuitively, what are we comparing? How do we build a tree out of Dirac masses in R^2? -- Page 8, NY Taxi Line 304: "The ground space is 2-dimensional space..." Isn't using the Manhattan distance more reasonable? The baseline is missing an existing unbalanced OT method on the ground metric. -- Trivial things Title: "on Tree(+s)" should be better Line 33: notati(+o)ns Line 63: "Kant(-o)rovich", since the rest of the paper uses "Kantrovich". It is a rare spelling of Kantorovich though. Line 270: Figure 3 should include the entire trajectories to show their (non-)linearity in the log scale and their relative positions at a small sample size. Line 287: "0.70 second(-s)"

Correctness: I haven't found major errors or weaknesses.

Clarity: The paper is generally well written.

Relation to Prior Work: Yes.

Reproducibility: Yes

Additional Feedback: ------------------------ After rebuttal: I have checked the rebuttal and other review. I am disappointed by the response to my questions and concerns because it dodges my points but the response to other comments is satisfactory in my opinion. Overall, I think the paper is marginally above the bar after incorporating the rebuttal into the new version. I will keep my original rating.


Review 2

Summary and Contributions: This paper deals with the unbalanced optimal transport problem. The authors define the gneralized Kantorovich Rubinstein (GKR) distance that allows mass creation and destruction. They propose an algorithm that solves the GKR problem in quasi-linear time on a tree metric. Well detailed numerical experiments are conducted on synthetic and real data.

Strengths: The novelty in this paper comes with a promosing algorithm for solving unbalanced OT problem. There is also a theoretical study of time complexities for the unbalanced OT from an algorithmic perspective. Structuring the unbalanced OT using tree metric is very interesting and seem to work very well in practice.

Weaknesses: Some comments are as follows: - To construct the cost tree metric, how we choose the weight function? - The authors proved only the triangle inequality for the GKR distance. I am wondering if GKR(mu, nu) = 0 iff mu = nu is automatically verified?

Correctness: All the experiments are well detailed and explained.

Clarity: Over all the paper is well written. Minor comments: - Ligne 145, the k-SAT problem is not defined before Hypothesis 1. - Ligne 161, by Indyk and Thaper

Relation to Prior Work: It would be interesting to compare the performance of the proposed algorithm to the generalized Sinkhorn algorithm to compute unbalanced OT studied in Chizat et al (2018).

Reproducibility: Yes

Additional Feedback:


Review 3

Summary and Contributions: This paper proposes the GKR distance, which encompasses many previous unbalanced OT problems. The authors show that important special cases of the GKR distance on Lp metrics cannot be computed in strongly subquadratic time under SETH. They then proposed a quasi-linear time algorithm to compute the GKR distance on tree metrics, which can be applied to large-scale problems.

Strengths: The GRK distance is interesting, and the proposed quasi-linear time algorithm for solving GRK distance is also fast and effective. The theoretical analysis of the proposed method seems to be sufficiently rigorous and provides some insights.

Weaknesses: (1) This paper proposes a general framework for unbalanced OT problems named GRK distance, one problem is how generality is reflected in the equation of the GRK distance. (2) A question is that if the proposed algorithm for GKR distance is also applicable to optimal partial transport and Kantrovich-Rubinstein distance. If not, what is the main reason why the proposed algorithm cannot be directly applied to; if it can be applied directly, what is the main advantage of GRK compared with optimal partial transport and Kantrovich-Rubinstein distance? (3) This paper does not explain why GKR distance is robust. (4) The reported evaluation results may not be sufficient to support the claim on the effectiveness of the GRK distance. It seems that there are no quantitative results in comparison with other distances and methods on some real-world tasks. There are many other datasets available on which the method can be further tested. It is lacking comparisons with some more recent methods during the past 1~2 years.

Correctness: The claims, method, and empirical methodology appear to be correct, but I didn’t check completely.

Clarity: The paper is pretty readable.

Relation to Prior Work: The relation between the GRK distance and some previous contributions is clearly discussed.

Reproducibility: Yes

Additional Feedback: (1) The formulas proj_1 \mu (A) = \mu (A \times X) and proj_2 \mu (B) = \mu (X \times B) on line 106 may or should be proj_1 \pi (A \times X) = \mu (A) and proj_2 \pi (X \times B) = \mu (B). (2) The formula (y,x) on line 112 may or should be written as (y,x) \in E. ------------------------------- After rebuttal: Thanks for the response. The rebuttal addressed part of my concerns, so I keep my rating.


Review 4

Summary and Contributions: The paper proposes a new formula to formalize the unbalanced OT problem, and many existing formulas for this problem can be treated as special cases of it. Further, the authors propose a tree-based algorithm that solves the unbalanced OT problem in quasi-linear time for large scale dataset.

Strengths: (1). The new formula for the unbalanced OT problem is powerful enough to cover a lot of other representations. (2). The performance of the algorithm seems to be promising in terms of the computation speed.

Weaknesses: (1). Basically, the paper is hard to follow, especially for the algorithm part (Sec. 5). Maybe the authors can use some figures and graphs to better illustrate their idea. In my opinion, it seems that the algorithm tries to recursively transport mass from the nodes of the source distribution to the nearby nodes in the target distribution. But the writing here makes me hard to grasp the basic ideas. (2) I am not sure if the algorithm can only solve the 2-dimensional unbalanced OT problem because the use of the quadtree structure. How to build the tree structure on high dimensional space? (3) If there is no intersection between the supports of the source and the target distributions, is the algorithm still feasible? (4) About the accuracy of the proposed algorithm: since both \lambda_d and \lambda_c are set to be constant, the GKR distance can be treated as solving a linear programming problem. Maybe the authors can experiment on a small toy dataset, then show the groundtruth GKR distance and the approximated distance by the proposed algorithm.

Correctness: (1) Proof of Thm.2: Between Line 505-506, it seems that there is a hidden assumption that c(x,y) >= 2\lambda. Why should this assumption work? (2) Is the OT_{tree} defined in Line 522 the same with |v(P)-v(Q)|_1 in [31]?

Clarity: (1) The illustration of the algorithm seems not clearly enough. Maybe the authors can use some figures and graphs to better illustrate their idea. (2) Some minor questions: a. What's the meaning of \omega in Line 151? b. Line 184: is there any reference for the computation complexity of the sinkhorn algorithm? c. Line 283: Figure 6 seems to be Figure 3?

Relation to Prior Work: Yes

Reproducibility: Yes

Additional Feedback: -------------------- After Rebuttal: I decide to change my score from 5 to 6, since the rebuttal solves some of my questions. The new illustration of the algorithms and experiments is convincing. But I am still skeptical about the performance of the algorithm, beyond the experiments, maybe some theoretical bound of the accuracy should also be given.

[Author Response · NeurIPS 2020]



All reviewers: We thank all reviewers for their positive feedback. We are encouraged that they find our work offers *a significant algorithmic breakthrough* (R1), *addresses an important open problem* (R1), *works very well in practice* (R2), and is *interesting* (R2, R3) and *promising* (R2, R4). ► *Generality.* Let us clarify that our algorithm and Theorem 3 are valid for many existing UOT problems. If one wants to compute the Kantrovich-Rubinstein (KR) distance, our algorithm computes it in $O(n \log^2 n)$ time by setting $\lambda_c = \lambda_d = \lambda$, and Theorem 3 provides an approximation guarantee. Note that any existing algorithms for the KR distance require at least $O(n^2)$ time. Likewise, if one wants to compute the Figalli's distance, our algorithm computes it efficiently by setting $\lambda_c(x) = \lambda_d(x) = d(x, \partial \mathcal{X})$. See also related work.

Reviewer #1: ► *Two hyper-parameters.* When two hyper-parameters are troublesome, we recommend to use the KR distance or optimal partial transport, which has only one hyper-parameter and can be computed efficiently by our algorithm. ► *Chicago Crime.* We compare the distributions of crime locations. For example, in some festival days, many crimes may happen in specific locations, and the number of crimes may suddenly increase/decrease. We can detect anomalies and clusters. ► *NY Taxi.* Existing methods for UOT are missing because *no existing methods can handle this dataset due to scalability*. Our method is the first UOT method that can handle million-scale datasets.

Reviewer #2: ► *Weight function.* The cost is the ground distance between the centers of regions of the quadtree. We will further clarify this in the camera-ready. ► *Metric Axiom.* Intuitively, when no mass is created or destructed, GKR is reduced to the standard OT, thus metric. When some mass is created or destructed, GKR is positive. Hence, $\mathrm{GKR}(\mu, \nu) = 0$ iff $\mu = \nu$ almost everywhere (under positivity conditions for $\lambda_c$ and $\lambda_d$). We will formally state this.

Reviewer #3: ► *(1) How generality is reflected.* Many existing OT problems are obtained by setting $\lambda_c$ and $\lambda_d$ appropriately. ► *(4) Recent methods.* The tree-sliced Wasserstein we used in the NY taxi dataset and Appendix E was published in NeurIPS 2019. That is a state-of-the art (standard, not unbalanced) OT method applicable to million-scale datasets. We also used the generalized Shinkhorn published in 2018 in the additional experiments below.

Reviewer #4: ► *(1) Hard to follow.* Our algorithm computes the GKR distance from leaf to root recursively. Figure 1 shows examples. Note that when we compute the transport in a parent node, the optimal assignments in the children subtrees are already computed recursively. When we merge two children, the dynamic programming determines the optimal transition (i.e., the optimal amount of mass that are transported to the left and right children). The proposed fast convolution algorithm speeds up this merge operation. We will provide more detailed descriptions and intuitions in the camera-ready. Furthermore, we will provide an open-sourced toolkit of our algorithm at GitHub. We believe that it will benefit many practitioners thanks to its fast computation. ► *(2) Beyond 2 dimensions.* The quadtree we used in the paper is *NOT* restricted to two dimensions. See [38, 31] for details. When the dimensions are high, clustering-based trees can be used [38]. To show this, we conduct additional experiments. First, we compute GKR for the Chicago Crime dataset with the additional time axis, using the *quadtree*. Each mass represents a crime in the 3-dimensional (longitude, latitude, time) space. We normalize each dimension so that each dimension has the same scale. Next, we compute GKR for the (unbalanced) Word Mover's

Figure 1: Two examples of DP computation.

Figure 2: Accuracy in high dimensional cases.

Distance of the Twitter dataset [38] using *clustering-based trees*. Each measure represents a sentence, and each mass represents a word embedded in a 300-dimensional space computed by a pre-trained language model. We compare our algorithm with the groundtruth GKR distance in the Euclidean space, as we did in the main paper. Figure 2 shows that our algorithm can compute high dimensional GKR. ► *(3) No intersection case.* Our algorithm *is* feasible even if there is no intersections. In that case, each leaf node contains only the source or target mass. ► *(4) LP formulation.* We reported the accuracy in Figure 4 in the original paper. There, we used *exact* computation for the Euclidean GKR using an LP-like solver. Specifically, we used a network flow algorithm, which solves OT problems exactly (i.e., match exactly with the LP solution) and is faster than general-purpose LP solvers. We will clarify it. ► *Proof of Thm.2.* There, we consider the case where $\lambda \le \delta/2 (\le c/2)$ (See L.505). See also the definitions of $\delta$ and $c$ in L.503-504. The opposite case (i.e., $\lambda < \delta/2$) is discussed in L.506-508. ► *Is $OT_{tree}$ the same as $|v(P) - v(Q)|_1$ in [31]?* Exactly.

Additional Experiments: We conducted experiments for the generalized Sinkhorn [16] with the same setting as Appendix E. We observed a similar tendency ($k$=0: 0.83, $k$=16: 0.37) to Tree GKR. The complete results are deferred to the camera-ready due to space limitation. Since the generalized Sinkhorn requires at least $O(n^2)$ time, its applicability is limited to thousand-scale datasets. Our algorithm is applicable to million-scale datasets keeping its performance. We also conducted document classification using Twitter dataset [38] and found that GKR improved the performance over the Word Mover's Distance (Accuracy: $0.719 \to 0.729$). The detailed results will be included in the camera-ready.

[Meta-Review · NeurIPS 2020]

Reviewers are marginally enthusiastic about this paper, which proposed a data tree-structure to simplify the solution to the original OT problem. In addition to the various issues reviewers raised, there are additional criticisms which the authors should consider in their revision. 1. An important relevant paper is not cited: https://papers.nips.cc/paper/6566-stochastic-optimization-for-large-scale-optimal-transport.pdf 2. The contributions include: (1) propose the tree-like structure to solve the unbalanced OT problem (which include standard OT as the special case) (2) prove that their time complexity is O(nlog^2n). However, the paper did not say anything about the accuracy, other than some experiments. The existing OT solvers needs O(n^2/\eps) complexity, where \eps is the additive error. The cost of OT is roughly on the order of n^2. Therefore, it is probably affordable to use \eps = \eps' n^2, which makes the complexity become O(1/\eps') , where \eps' is the relative error. In their experiments, the relative error is about 0.05, which is fairly easy to achieve using existing OT solver.